# communications
# engineering

https://doi.org/10.1038/s44172-023-00062-7　　OPEN
# Heavy metal removal from coal fly ash for low carbon footprint cement

Bing Deng [1,9✉], Wei Meng[2,9], Paul A. Advincula[1], Lucas Eddy [1,3], Mine G. Ucak-Astarlioglu[4], Kevin M. Wyss[1], Weiyin Chen[1], Robert A. Carter[1], Gang Li[1], Yi Cheng [1], Satish Nagarajaiah [2,5,6,7✉] & James M. Tour [1,5,6,8✉]

Development of cementitious materials with low carbon footprint is critical for greenhouse gas mitigation. Coal fly ash (CFA) is an attractive diluent additive in cement due to its widespread availability and ultralow cost, but the heavy metals in CFA could leach out over time. Traditional acid washing processes for heavy metal removal suffer from high chemical consumption and high-volume wastewater streams. Here, we report a rapid and water-free process based on flash Joule heating (FJH) for heavy metals removal from CFA. The FJH process ramps the temperature to ~3000 °C within one second by an electric pulse, enabling the evaporative removal of heavy metals with efficiencies of 70–90% for arsenic, cadmium, cobalt, nickel, and lead. The purified CFA is partially substituted in Portland cement, showing enhanced strength and less heavy metal leakage under acid leaching. Techno-economic analysis shows that the process is energy-efficient with the cost of ~$21 ton$^{-1}$ in electrical energy. Life cycle analysis reveals the reuse of CFA in cement reduces greenhouse gas emissions by ~30% and heavy metal emissions by ~41%, while the energy consumption is balanced, when compared to landfilling. The FJH strategy also works for decontamination of other industrial wastes such as bauxite residue.

[1] Department of Chemistry, Rice University, 6100 Main Street, Houston, TX 77005, USA. [2] Department of Civil and Environmental Engineering, Rice University, 6100 Main Street, Houston, TX 77005, USA. [3] Applied Physics Program, Rice University, 6100 Main Street, Houston, TX 77005, USA. [4] Geotechnical and Structures Laboratory, U.S. Army Engineer Research and Development Center, Vicksburg, MS 39180-6199, USA. [5] Smalley-Curl Institute, Rice University, 6100 Main Street, Houston, TX 77005, USA. [6] Department of Materials Science and NanoEngineering, Rice University, 6100 Main Street, Houston, TX 77005, USA. [7] Department of Mechanical Engineering, Rice University, 6100 Main Street, Houston, TX 77005, USA. [8] NanoCarbon Center and the Welch Institute for Advanced Materials, Rice University, 6100 Main Street, Houston, TX 77005, USA. [9]These authors contributed equally: Bing Deng, Wei Meng. ✉email: bingdeng@rice.edu; satish.nagarajaiah@rice.edu; tour@rice.edu

The growing global demand for materials continuously increases greenhouse gas (GHG) emissions[1]. Building materials are the third-largest source of anthropogenic carbon dioxide ($CO_2$) emission[2]. For example, global $CO_2$ emissions of cement production are ~1.5 Gt annually, representing ~8% of the total global GHG emissions[2,3]. Hence, the cement industry is an important sector for GHG emission mitigation strategies[4], and there is renewed interest in alternative raw materials[5] with lower production emissions to replace or partially substitute the highest volume building material, ordinary Portland cement (OPC).

Among the alternative cementitious materials[5], coal fly ash (CFA) has been extensively investigated and utilized worldwide[6]. CFA is the predominantly inorganic residue of coal combustion in power plant furnaces, with an annual worldwide production of ~750 million tons[7,8]. CFA is primarily silicon (Si), aluminum (Al), iron (Fe), and calcium (Ca) oxides while containing smaller amounts of heavy metals, such as cadmium (Cd), cobalt (Co), copper (Cu), nickel (Ni), lead (Pb), and mercury (Hg)[9,10]. Hence, CFA is classified as hazardous waste in some countries if the toxic substances exceed limits; other countries regulate it as non-hazardous with special regulations[11]. The improper disposal of CFA has become an environmental concern because of potential water, soil, and air pollution[6,12]. As a result, reuse, rather than disposal or landfilling, of CFA is sought for both economic and environmental reasons[2,7]. CFA with high calcium content has considerable cementitious properties[13], making it useful for OPC dilution[14,15]. However, the leachability of heavy metals from CFA[16,17], when it is in contact with aqueous environments such as rainwater, has impeded its applications in cement[18,19]. Therefore, the removal of toxic elements from CFA is necessary prior to landfilling or secondary use. The present methods for heavy metal removal mostly rely on the acid washing process, including the use of inorganic[20] or organic acids[21], both of which suffer from the consumption of chemicals and generation of large wastewater streams that reduce the value of economic incentives and result in secondary pollution[17,20].

Recently, highly efficient, short-burst electric heating is emerging as a high-temperature technology for materials production[22–28] and solid waste management[29]. Chen et al. first reported the rapid Joule heating for the ultrafast synthesis of nanoparticles in reduced graphene oxide films[27]. The carbothermic shock was then widely applied for various nanomaterial syntheses[30], including silicon nanoparticles[31], high-entropy alloy nanoparticles[28], and single-atom catalysts[32]. The FJH process has been used to convert carbon-containing sources into flash graphene[33]. In addition to the functional materials synthesis capability[34–36], the FJH process has been demonstrated to be an efficient method for sustainable management of carbon-rich wastes, such as consumer plastic[37,38], rubber[39], end-of-life vehicle waste[40], and asphaltenes[41]. With the ultrahigh temperature reaching ≥3000 °C and ultrafast process lasting ≤1 s, the FJH method enables the evaporative separation of precious metals from electronic wastes for urban mining[42], the activation of industrial wastes for high-yield rare earth elements recovery[43], the recycling of photovoltaic silicon waste[44], and recovery of lithium-ion batteries graphite anodes[45–47] and cathodes[48].

Here, we report that the FJH strategy can be applied to remove heavy metals rapidly and efficiently from CFA. The FJH process ramps the temperature to ~3000 °C within 1 s, enabling the evaporative removal of various heavy metals from CFA with efficiencies of 70–90% for As, Cd, Co, Ni, and Pb within a single FJH treatment. The removal efficiencies are further increased by repeating the 1 s FJH pulse. The FJH strategy works for CFA regardless of the types (class F and class C) and the geographical origins. We substitute the FJH-purified CFA for up to 30 wt%

OPC, and the resulting composite shows an enhanced compressive strength of ~51% and modulus of ~28% compared to these from pure OPC. Acid rain leaching experiments show that the cement made from purified CFA exhibits the least heavy metal leakage. Due to the rapid treatment process and ultrafast heating and cooling rates, the FJH process is highly energy-efficient with an estimated cost of ~\$21 ton$^{-1}$ in electrical energy for CFA treatment. A life cycle analysis (LCA) was conducted to assess the environmental impacts of the reuse of CFA in cement and CFA landfilling. The LCA indicates that CFA substitution in cement strategy effectively reduces GHG emissions by ~30% and heavy metal emissions by ~41%. The same FJH strategy was further applied to the purification of large-scale bauxite residue (red mud), demonstrating the generality of the process for solid waste decontamination and valorization.

## Results

**Heavy metals in CFA.** Based on chemical composition, CFA is categorized into class F CFA (CFA-F) and class C CFA (CFA-C). While both contain major components of $SiO_2$, $Al_2O_3$, and $Fe_2O_3$, CFA-C has a high abundance of CaO[49]. In this work, the CFA-F was collected from the Appalachian Basin (App) and CFA-C from Powder River Basin (PRB), both in the United States. While CFA is mostly composed of glassy phases produced during the coal burning process[50], the crystalline components mainly include quartz ($SiO_2$) and mullite (aluminum silicate, $3Al_2O_3·2\text{-}SiO_2$), according to X-ray diffraction (XRD) analyses (Fig. 1a). In addition to the Ca enrichment in CFA-C, elemental analysis by X-ray photoelectron spectroscopy (XPS) also shows an abundance of carbon in CFA-F (Fig. 1b), which might be from the incomplete combustion of coal. The morphology of the CFA was characterized by scanning electron microscopy (SEM). The particle size of CFA-C is ~1–10 μm (Fig. 1c), while the CFA-F is ~1–8 μm (Fig. 1d).

The CFA samples were digested by acid (see details in the "Methods" section), and the trace heavy metal contents were measured by inductively coupled plasma mass spectrometry (ICP-MS). It is found that the As, Cd, Co, Ni, and Pb exist in the CFA, with As, 59.7 ± 3.3 ppm; Cd, 0.76 ± 0.36 ppm; Co, 15.9 ± 3.8 ppm; Ni, 36.6 ± 8.4 ppm; and Pb, 22.8 ± 1.7 ppm for CFA-C (Fig. 1e); and As, 88.6 ± 43.0 ppm; Cd, 0.62 ± 0.08 ppm; Co, 18.7 ± 5.5 ppm; Ni, 43.5 ± 13.5 ppm; and Pb, 28.3 ± 8.6 ppm for CFA-F (Fig. 1f). It is intriguing that the heavy metal content in the CFA-F and CFA-C samples are similar even though they are different types and from different geological origins.

**Removal of heavy metals in CFA by FJH.** In a typical FJH experiment, the CFA was mixed with carbon black (CB), ~30 wt %, which serves as the conductive additive. The mixture was loaded into a quartz tube, which was connected to a capacitor bank (Fig. 2a). The electric diagram and setup of the FJH system are shown in Supplementary Fig. 1. The resistance of the sample was controlled by the compressive force of the two electrodes; in most of the trials, the resistance was fixed to be ~1 Ω (Supplementary Table 1). Too high or too low resistances result in inferior FJH reactions: a resistance too high does not afford high enough current for Joule heating, and a resistance too low does not generate enough heat. The detailed conditions for FJH are shown in Supplementary Table 1. For a typical discharge with a voltage of 120 V and discharge time of 1 s, the current passing through the sample was recorded to be ~120 A at its maximum (Fig. 2b). The fluctuation of the current curve is ascribed to the changing of sample resistance due to degassing or intrinsic temperature-dependent resistivity. The capacitor discharge produces a sample temperature of up to ~3000 °C in 5 ms (Fig. 2c),

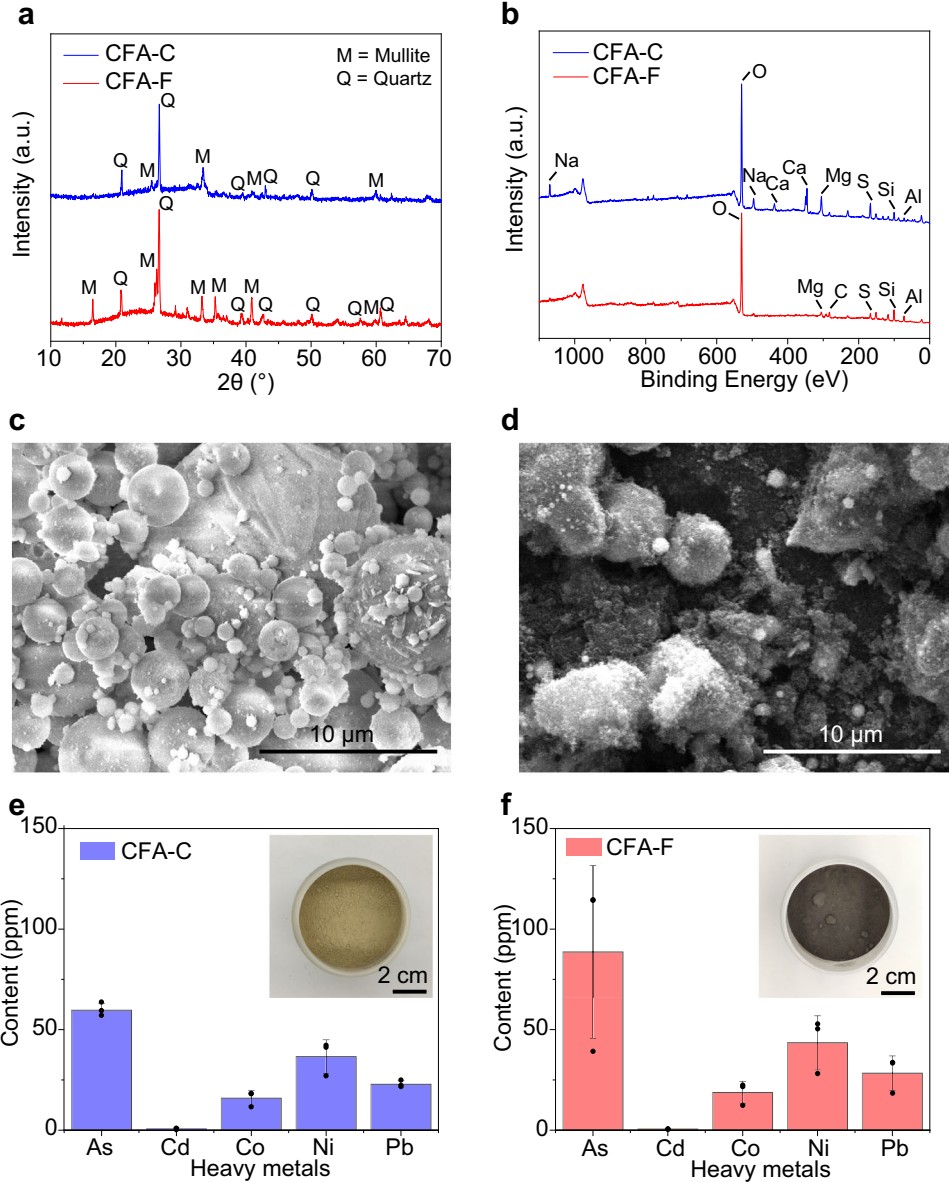

**Fig. 1 Characterization of CFA. a** XRD patterns of class C coal fly ash (CFA-C) and class F coal fly ash (CFA-F). Mullite (PDF 15-0776) and quartz (PDF 33-1161) are used as references. **b** XPS full spectra of CFA-C and CFA-F. **c** SEM image of CFA-C. **d** SEM image of CFA-F. **e** Heavy metal content in CFA-C. Inset, picture of CFA-C. **f** Heavy metal content in CFA-F. Inset, picture of CFA-F. The error bars in **e** and **f** denote the standard deviation where $n = 3$.

followed by rapid cooling. The temperature keeps changing during the FJH process due to the sample resistance and current variation. Such a high temperature enables the vaporization of the heavy metals including Cd, As, Pb, Co, and Ni, according to their vapor pressure-temperature relationships[51] (Fig. 2d). In contrast, the CB conductive additives are transformed to graphite-like carbon[33], which does not sublime until ~3600 °C[52].

The heavy metal content in the residual solid after FJH were measured by ICP-MS, and their removal efficiencies were calculated (Supplementary Note 1). The heavy metal contents in the CB are 2–15% of those in CFA (Supplementary Fig. 2, Supplementary Discussion 1), a significant amount. Thus, in the calculation of the removal efficiencies, the combined total heavy metal content in CFA and CB was used as the baseline. A series of FJH voltages ranging from 60 to 150 V was applied (Fig. 2e) to purify CFA-F. The heavy metal removal efficiencies increased from 60 to 120 V, which could be ascribed to a higher sample temperature produced by the higher FJH voltage[43]. With an FJH

voltage of 120 V, the heavy metal removal efficiencies were 70–90% by one FJH pulse (Fig. 2f). Further increasing the FJH voltage to 150 V did not increase the removal efficiencies (Fig. 2e), which could be due to inhomogeneous heating under excessive energy input. The evaporated heavy metals were deposited onto the sidewall of the quartz tube reactor or inside the sealed chamber (Supplementary Fig. 1b, d), in avoidance of emission to the environment.

The CFA-C was also used as the starting material. Under the FJH voltage of 120 V, the removal efficiencies achieve 40–80% for the representative heavy metals (Supplementary Fig. 3). Generally, physicochemical adsorption methods rely on the capacity of sorbents, thus the heavy metal removal capacity is limited[53]. In contrast, the FJH process has no capacity limit due to its evaporative removal feature. With multiple FJH pulse reactions, we demonstrated that the removal efficiencies of heavy metals can be increased to >75% for Ni and >85% for Cd, Co, and Pb for CFA-C (Supplementary Fig. 4).

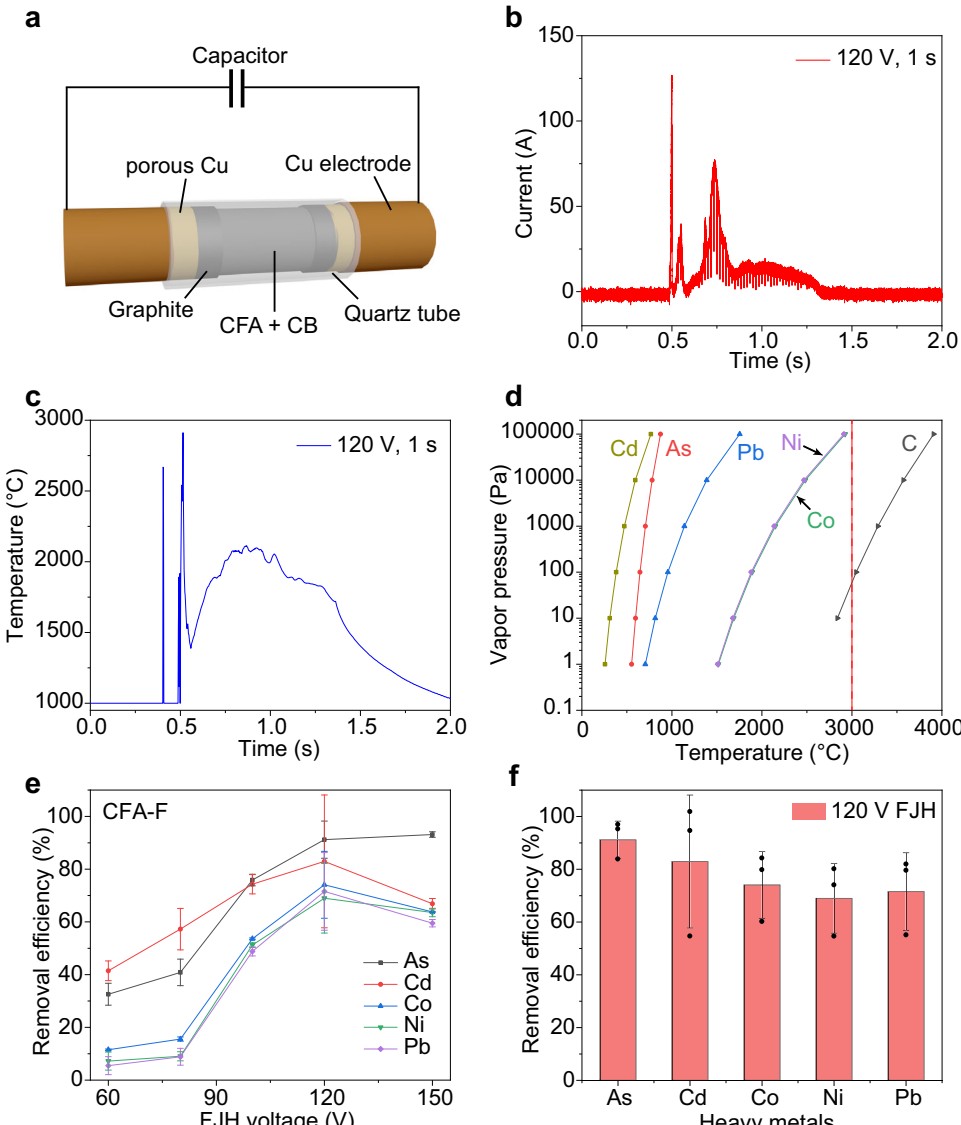

**Fig. 2 Removal of heavy metals in coal fly ash (CFA) by flash Joule heating (FJH). a** Schematic of the FJH process to removal heavy metals in CFA. CB, carbon black. **b** Current curve with FJH voltage of 120 V for 1 s. **c** Realtime temperature curve with FJH voltage of 120 V for 1 s. **d** Vapor pressure–temperature relationship of representative heavy metals and carbon. The vapor pressure values are from CRC Handbook of Chemistry and Physics[50]. The dashed line denotes the temperature of 3000 °C. **e** Removal efficiencies of heavy metals from class F coal fly ash (CFA-F) vs. the FJH voltage. **f** Single FJH removal efficiencies of heavy metals from CFA-F at the FJH voltage of 120 V. The error bars in **e** and **f** denote the standard deviation where $n = 3$.

The heavy metals in CFA are present in the oxidized or natural ore forms, according to previous studies[10,54,55]. Depending on the reactivity and thermal stability, the heavy metal species could evaporate in the natural form, or they could undergo thermal decomposition or carbothermic reduction to other compounds or elemental metals and then evaporate. In any case, the ultrahigh temperature by the FJH process would enable the chemical conversion and evaporation of the heavy metal species, which usually take place far below 3000 °C (Supplementary Fig. 5, Supplementary Discussion 2).

The optimized mass ratio of CFA and CB was ~2:1 (Supplementary Table 1), where the resistance of the sample was ~1 Ω (Supplementary Fig. 6, Supplementary Discussion 3). Other than CB, other inexpensive carbon could also be used as the conductive additive. For example, by using metallurgical coke (metcoke) as the conductive additive, the heavy metal removal efficiencies from CFA-F are 40–90% with one FJH pulse at a voltage of 120 V

(Supplementary Note 1, Supplementary Fig. 7, Supplementary Discussion 4). Moreover, plastic pyrolysis ash (Plastic Ash), the byproduct of plastic pyrolysis[56], was also used as the conductive additive (Supplementary Fig. 8, Supplementary Discussion 5). The removal efficiencies were >60% in a single FJH pulse. Considering the low or negative value of pyrolysis ash[57], the material cost of the FJH purification process is presumed to be near zero. Under the same FJH parameters, the removal efficiencies using metcoke or Plastic Ash are somewhat smaller than those using CB as the additive (Fig. 2f). This might be due to the better conductivity and smaller particle size of CB, which permits a higher temperature and a more uniform heating. We presume that this could be compensated by increasing the FJH pulses (Supplementary Fig. 4) when metcoke or Plastic Ash is used as the conductive additive.

After the FJH treatment process, there is considerable residual carbon content in the remaining solid. The residual carbon could be removed by calcination, which will be discussed later. In

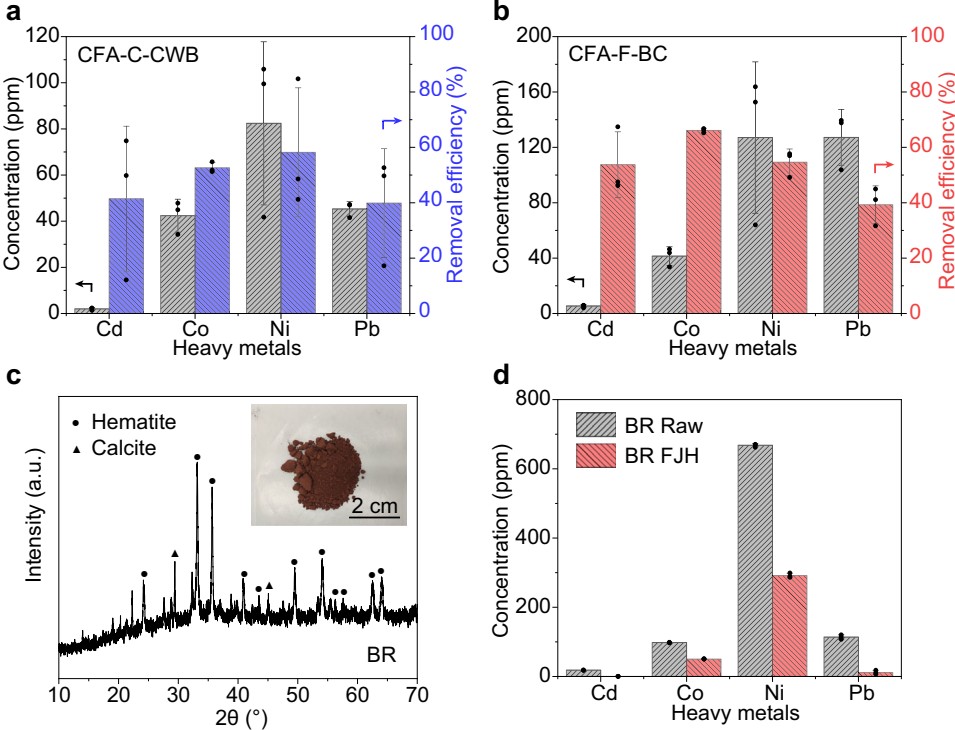

**Fig. 3 Generality of the flash Joule heating (FJH) process for heavy metal removal. a** Heavy metal content in class C coal fly ash collected from Charah White Bluff (CFA-C-CWB) and the removal efficiencies by FJH at 120 V for 1 s. **b** Heavy metal content in class F coal fly ash collected from Boral Cumberland (CFA-F-BC) and the removal efficiencies by FJH at 120 V for 1 s. **c** XRD pattern of bauxite residue (BR). Inset, the picture of BR. Hematite (PDF 02-0919) and calcite (PDF 47-1743) are used as references. **d** Heavy metal content in BR raw materials and the BR after a single FJH process. The error bars in **a**, **b**, and **d** denote the standard deviation where $n = 3$.

addition, based on the particle size and density differences between CFA and carbon, it is feasible to separate residual carbon from CFA using physical processes. By using metcoke as an example, we demonstrated the separation of purified CFA and metcoke by sieving (Supplementary Figs. 9a–d, Supplementary Discussion 6) with a metcoke recovery yield of ~92%. The recovered metcoke could be reused as the conductive additive for further purification of CFA (Supplementary Figs. 9e, f), which reduced the FJH purification cost. After the sieving separation process, the residual carbon content in the treated CFA was reduced to ~3% (Supplementary Fig. 10). The residual carbon could be completely removed by calcination in air, which will be discussed later. The choice of appropriate carbon removal approaches would depend on the landfilling or applications of the purified CFA.

In addition to the trace heavy metals, the main composition of the residual solid was characterized. The crystal components remain quartz and mullite after FJH for both CFA-C and CFA-F, according to their XRD patterns (Supplementary Fig. 11). Since amorphous phases that are unable to be detected by XRD generally account for >60% of CFA composition[58], X-ray fluorescence (XRF) was also used to quantify the composition change. It is found that the main composition, including various oxides, underwent little change after the FJH process (Supplementary Fig. 12), which is ascribed to the ultrafast heating and cooling rates and very short heating duration of the FJH process.

**Generality of the FJH process for heavy metals removal**. The above results show that the FJH works for CFA of different classes. CFA from different geological origins could have significant variations in their trace heavy metal content. To demonstrate the generality of the FJH process for heavy metals

removal, CFA from different sources were used as feedstocks, including CFA-C collected from Charah White Bluff (termed CFA-C-CWB), and CFA-F from Boral Cumberland (termed CFA-F-BC), both in the United States. The main composition of CFA-F-BC and CFA-C-CWB are quartz and mullite (Supplementary Figs. 13, 14), similar to those from App and PRB. The major heavy metals in CFA-C-CWB and CFA-F-BC were Cd, Co, Ni, and Pb (Fig. 3a, b). At a FJH voltage of 120 V (Supplementary Table 1), the heavy metal removal efficiencies were 40–60% for CFA-C-CWB (Fig. 3a) and 40–70% for CFA-F-BC (Fig. 3b) using one FJH pulse, demonstrating that the FJH process is a versatile process for CFA from different geological origins.

The FJH purification process can be further extended to other large-scale solid wastes, such as bauxite residue (BR), the by-product of the Bayer process for alumina production[59]. As one of the most abundant industrial wastes, BR has a production rate of 150 million tons per year in addition to the 3 billion tons already accumulated[60]. BR contains a significant content of heavy metals[61]. BR is a red powder in its dry form (inset in Fig. 3c), and XRD shows the major components of hematite and calcite (Fig. 3c). Similar to CFA, the BR was mixed with CB and the FJH process was conducted (Supplementary Table 1, Supplementary Fig. 15, Supplementary Discussion 7). The abundant heavy metals in BR include Cd, Co, Ni, and Pb (Fig. 3d). After the FJH process, the heavy metal concentrations were significantly reduced (Fig. 3d), demonstrating the generality of the FJH process for waste decontamination.

**Application of purified CFA-C in cement composites**. The CFA-C with a high content of CaO (~22 wt%, Supplementary Fig. 12) could be used as cementitious materials[14,15]. After the FJH reaction, the CFA-C contains ~10 wt% residual carbon

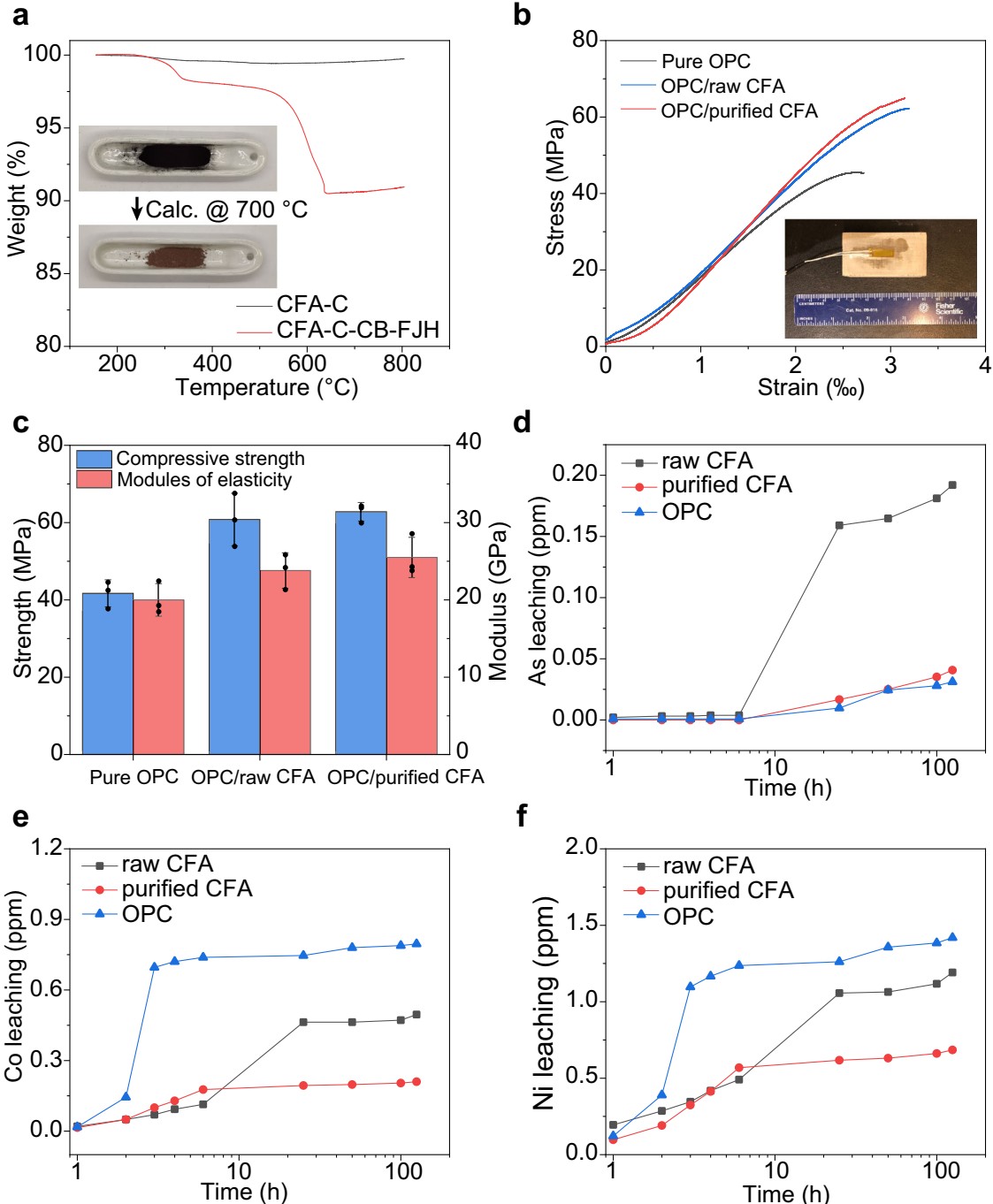

**Fig. 4 Application of purified class C coal fly ash (CFA-C) in cement composites. a** TGA curve of CFA-C raw materials and the mixture of CFA-C and carbon black (CB) after flash Joule heating (FJH). Inset, picture of the mixture of CFA-C and residual carbon, and the purified CFA-C after calcination in air at 700 °C for 1 h. **b** Stress–strain curves of the pure ordinary Portland cement (Pure OPC), the OPC substituted with 30 wt% raw CFA-C (OPC/raw CFA), and the OPC substituted with 30 wt% purified CFA-C (OPC/purified CFA). Inset, the picture of a cement sample for mechanical properties measurement. **c** Compressive strengths and moduli of elasticity statistics of the Pure OPC, OPC/raw CFA, and OPC/purified CFA. The error bars denote the standard deviation where $n = 3$. **d–f** Accumulated heavy metals content leached from the raw CFA cement, purified CFA cement, and OPC cement for **d** As, **e** Co, and **f** Ni.

according to the thermogravimetric analysis (TGA) (Fig. 4a). Prior to use, the residual carbon was removed by calcination in air at 700 °C for 1 h (inset in Fig. 4a). We note that the calcination process does not change the main composition of CFA-C according to the XRF analysis (Supplementary Fig. 12). In addition, the FJH and calcination processes did not substantially change the microscopic morphology of the CFA-C materials (Supplementary Fig. 16).

The purified CFA-C was used to substitute 30 wt% of OPC for cement composite (termed OPC/purified CFA), and pure OPC cement (termed Pure OPC) and the raw CFA-C substituted to 30 wt% of OPC (termed OPC/raw CFA) were tested as controls (see details in the "Methods" section). After only one day of curing, the compressive strength and modulus of elasticity of the OPC/purified CFA composites achieved 33.4 MPa and 15.5 GPa, respectively, much greater than those of Pure OPC at 20.6 MPa

and 8.9 GPa, respectively (Supplementary Fig. 17, Supplementary Discussion 8, Supplementary Table 2). The representative stress–strain curves of the Pure OPC, the OPC/raw CFA, and OPC/purified CFA after curing for 28 days are shown in Fig. 4b. The compressive strength of the OPC/purified CFA is 62.8 ± 2.4 MPa, exhibiting ~51% increase compared with that of the Pure OPC (Fig. 4c, Supplementary Table 2); the modulus of elasticity of OPC/purified CFA is 25.5 ± 2.6 GPa, which is ~28% greater than that of the Pure OPC (Fig. 4c, Supplementary Table 2). In addition, the performance of OPC/purified CFA is comparable to those of the OPC/raw CFA. This is probably because the composition (Supplementary Fig. 12) and particle size and morphology (Supplementary Fig. 16) of the CFA remained similar after the FJH purification process, except for the lowered heavy metal content.

Our group has previously demonstrated that appropriate loading (~0.15 wt%) of flash graphene into OPC promotes the composite's compressive strength[33,38,39]. However, the as-obtained CFA-C/CB after FJH has a high residual carbon content of ~10 wt%. To avoid a calcination process to remove the residual carbon, we further used the as-obtained purified CFA-C/CB to substitute 5 wt% OPC, thus with a nominal carbon content of ~0.5 wt%. In this case, the purified CFA-C/CB substituted cement showed a performance similar to that of raw CFA-C (Supplementary Fig. 18). In addition to CFA-C, the purified CFA-F was also used to substitute 30 wt% OPC, which exhibits comparable performance compared to pure OPC (Supplementary Fig. 19).

To mimic the acid rain leaching conditions, three cement pastes made from pure OPC, raw CFA-C, and purified CFA-C were put into a pH 4 $HNO_3$ solution, and the accumulated heavy metals were measured from 1 to 125 h (see details in the "Methods" section). As shown in Fig. 4d, the raw CFA-C has a serious As leaching up to ~0.2 ppm; in contrast, the purified CFA-C shows much less As leaching, comparable to pure OPC (Fig. 4d). In addition, the pure OPC exhibits the most severe Co (Fig. 4e) and Ni (Fig. 4f) leaching, while both are substantially lessened for the purified CFA-C. Hence, the purified CFA-C could be more environmentally friendly than OPC when considering heavy metal leakage, serving as another incentive for the application of purified CFA in real-world applications.

**Techno-economic analysis and life cycle analysis**. The FJH process for CFA purification has good scalability. The evaporative removal of the heavy metals depends on the maximum temperature during FJH; hence, maintaining a constant temperature is key for scaling up the FJH process. We first conducted a theoretical analysis of the FJH process, which demonstrates that the sample mass per batch could be increased by linearly increasing the FJH voltage or the total capacitance (Supplementary Note 2, scaling rule of the FJH process by theoretical analysis). By building an FJH system with the large capacitance of $C = 0.624$ F, we demonstrated the sample mass up to 3 g per batch (Supplementary Fig. 20, Supplementary Note 2, scaling up to gram scale per batch). By using a 3D printed automation system, we have realized a production rate of >10 kg day$^{-1}$ of flash graphene in our research laboratory (Supplementary Note 2, scaling up to kg-scale in our research lab). The FJH process could be integrated with some well-established scaling-up techniques for continuous processing (Supplementary Fig. 21, Supplementary Note 2, the conceptual design of the continuous FJH process). The FJH process is presently undergoing commercial scaleup for flash graphene synthesis, *en route* to a productivity of >100 ton per day by 2024[33]. Even though it is designed for flash graphene synthesis, the equipment and processes could be leveraged for the heavy metal removal process (Supplementary Note 2).

The energy consumption and cost of the FJH purification process has been estimated. Joule heating is a highly efficient technique with a coefficient of performance of nearing 1.0 since almost all the electrical energy directly targets sample heating. This is in striking contrast to a traditional furnace that relies on thermal conduction to heat the sample, leading to reduced energy efficiency. Because of the ultrafast heating and cooling rates and the short processing time within 1 s, the FJH process for heavy metal removal from CFA has an estimated electricity consumption of ~532 kWh ton$^{-1}$, or $21 ton$^{-1}$ using an industrial electricity rate of Texas, the US (Supplementary Note 3). The materials cost could be minimized by recovering and reusing the conductive additives, or by using conductive additives with low-cost or negatively valued carbon such as Plastic Ash.

A comparative cradle-to-gate life cycle analysis (LCA) was conducted to examine the environmental impact and energy demand resulting from the disposal of CFA as compared to the reuse of unpurified or purified CFA as alternative cementitious materials. The cradle-to-gate LCA scope, goal, scenarios, boundary conditions, and inventory (listing all inputs, outputs, and processes) considered in our LCA, following ISO guidelines, are included in Methods, Supplementary Note 4, and Supplementary Tables 3–7. Four scenarios were considered in this study (Fig. 5a, b, Supplementary Fig. 22), namely, Landfilling (pure OPC as cement for service life, and CFA and Plastic Ash being landfilled), Direct Substitution (OPC–raw CFA composite as cement for service life, and Plastic Ash being landfilled), FJH-Separation-Substitution (CFA purified by FJH followed by the removal of residual carbon by separation, and OPC-purified CFA composite as cement for service life), and FJH-Substitution (CFA purified by FJH without the removal of residual carbon, and OPC-purified CFA–Plastic Ash composite as cement for service life).

Two environmental impacts, heavy metal emissions and GHG emissions, and energy consumption were analyzed. First, as expected, the FJH-Separation-Substitution scenario has the least heavy metal emissions (Fig. 5c, Supplementary Table 5), demonstrating ~41% reduction compared to the Landfilling scenario (Supplementary Fig. 23a). The Direct Substitution scenario also exhibits a ~22% reduction in heavy metal emissions because of the lower heavy metal leakage of CFA than that of OPC (Fig. 4d–f). Second, for the GHG emission, the Landfilling scenario has a tremendous $CO_2$ emission of 854 kg per ton of cementitious materials, the vast majority of which is from the production of OPC (Fig. 5d, Supplementary Table 6). All other scenarios with CFA partial substitution for OPC show GHG emission reduction, i.e., ~30% for Direct Substitution, ~30% for FJH-Calcination-Substitution, and ~5% for FJH-Substitution compared to Landfilling (Supplementary Fig. 23b). Last, the Direct Substitution scenario has the least energy consumption at 3310 MJ ton$^{-1}$, representing ~29% reduction when compared to that of Landfilling (Fig. 5e, Supplementary Table 7, Supplementary Fig. 23c). The scenarios with the FJH purification process demonstrate a slight decrease in energy consumption of ~1% for FJH-Separation-Substitution, and a slight increase of ~2% for FJH-Substitution (Supplementary Fig. 23c) thanks to the highly energy-efficient FJH process, as we discussed above. Hence, the energy consumption of the FJH process is balanced by the reduced consumption of OPC.

## Discussion
We compare the FJH with existing methods[62] for heavy metal removal from CFA (Supplementary Table 8), including bioleaching[63], leaching using inorganic acid[64] or organic acid[65], chemical extraction by alkaline leachates[66] or chelating agents[66].

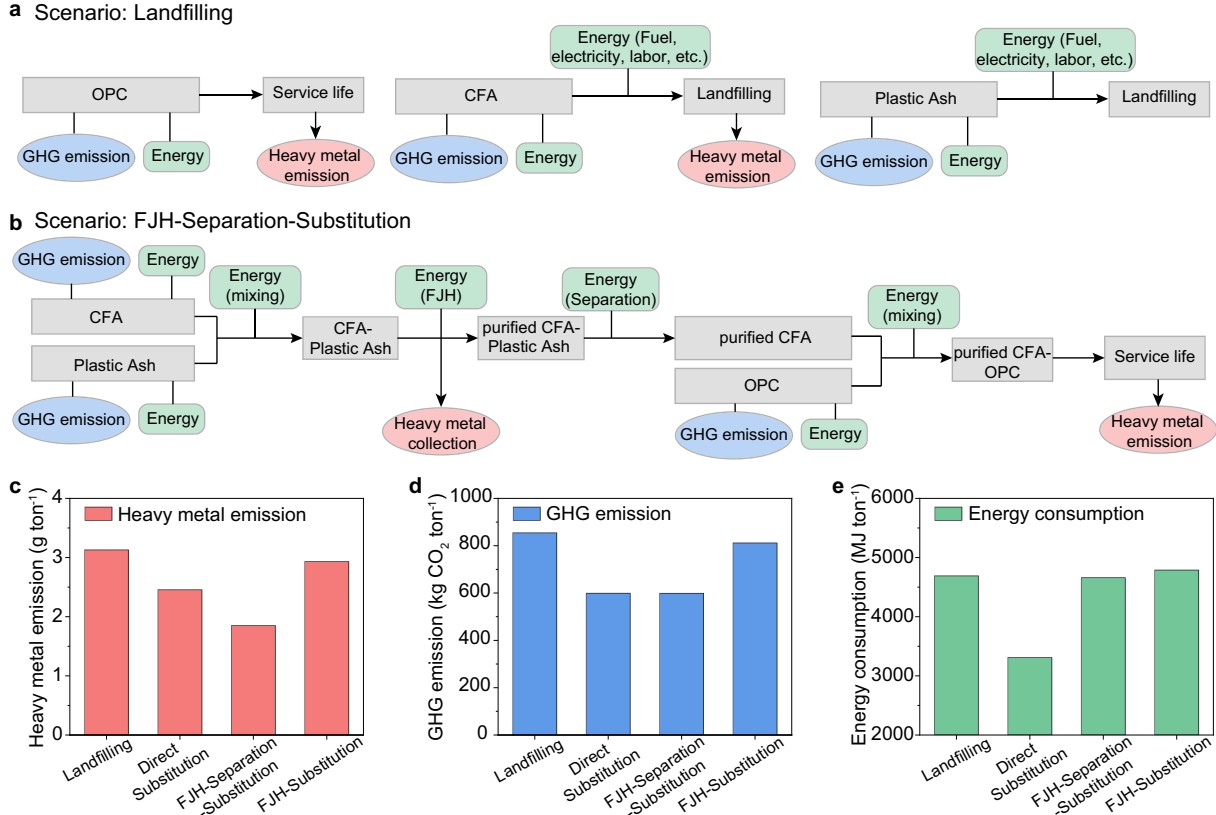

**Fig. 5 LCA for the reuse of coal fly ash (CFA) in cement. a** Flow chart representation and boundary conditions for the LCA scenario of Landfilling. OPC ordinary Portland cement. **b** Flow chart representation and boundary conditions for the LCA scenario of FJH-Separation-Substitution. FJH flash Joule heating, GHG greenhouse gas. **c** Comparison of heavy metal emissions. **d** Comparison of GHG emissions. **e** Comparison of energy consumptions.

The FJH is a water-free process, while the bio- or chemical processes consume large amounts of water (Supplementary Fig. 24a). The chemical leaching methods use large amounts of chemicals such as acid, base, and chelating agents, thus the materials cost is higher than the bioleaching and FJH processes (Supplementary Fig. 24b). The FJH process affords rapid treatment within seconds, much faster than the sluggish leaching processes (Supplementary Fig. 24c). Lastly, the heavy metal removal efficiency of the FJH process resembles that of inorganic acid leaching efficiency, both of which are superior to other processes (Supplementary Fig. 24d).

## Conclusion

In conclusion, we report an FJH strategy for the removal of toxic heavy metals from CFA with high removal efficiencies of 70–90% by a single one-second FJH pulse. We prepared the purified CFA-derived cement substituted to 30 wt% OPC, which showed an enhanced strength of ~51% and modulus of ~28% compared to that of pure OPC. The simulated acid rain leaching experiments show that the cement from purified CFA exhibits much less heavy metal leakage than raw CFA, and is even better than pure OPC. The cradle-to-gate LCA revealed that the reuse of CFA in cement could reduce heavy metal emissions by 41% and GHG emissions by 30% compared to the current waste management practice (landfilling). Due to the rapid treatment process and ultrafast heating and cooling rate, the FJH process is highly energy-efficient with an estimated electrical cost of ~\$21 ton$^{-1}$ for CFA treatment. The FJH strategy is also applicable for decontaminating other wastes like BR. The ongoing commercial scaleup of the FJH process makes it appealing in the decontamination and valorization of large-scale industrial wastes.

## Methods

**Material**. CB (Cabot, Black Pearls 2000, average particle size ~10 nm), metcoke (SunCoke Energy, average particle size <150 μm), and plastic pyrolysis ash (Shangqiu Zhongming Eco-Friendly Equipment Co., Ltd in Shangqiu City, Henan, China) were used as the conductive additives. The CFA-C and CFA-F samples were collected from PRB and App, respectively, both in the United States, and kindly provided to our laboratory; see Acknowledgement. The CFA-C-CWB was collected from Charah White Bluff, and the CFA-F-BC from Boral Cumberland, both in the United States. The BR sample was collected from MYTILINEOS S.A. in Greece and kindly provided to our laboratory; see Acknowledgement.

**FJH system and the heavy metal removal process**. The FJH system was described in our previous publications[33,42]. The electrical diagram is shown in Supplementary Fig. 1a. The mixture of CFA and CB ($w:w = 2:1$) with mass of ~150 mg was loaded into a quartz tube with an inner diameter of 8 mm and outer diameter of 12 mm. Graphite rods in contact with the sample and porous Cu electrodes were used as the electrodes in both sides of the quartz tube. The graphite rods were loosely fitted in the quartz tube to permit outgassing. The tube was then loaded on the reaction stage (Supplementary Fig. 1b) and connected to the FJH system (Supplementary Fig. 1c). The reaction stage was placed in a plastic vacuum desiccator with a mild vacuum of ~20 mm Hg to facilitate the degassing (Supplementary Fig. 1d) after the reaction. The resistance of the sample was controlled by compressing the electrodes. A capacitor bank with a total capacitance of 60 mF was charged by a DC supply that can reach voltages up to 450 V. A relay with programmable ms-level delay time was used to control the discharge time. The discharging of the capacitor brings the sample to a high temperature. The detailed conditions for the FJH reactions are listed in Supplementary Table 1. After the FJH reaction, the apparatus was rapidly cooled to room temperature. The heavy metal content in the CFA was measured before and after the FJH process to determine the removal efficiencies of contaminants. CAUTION: There is a risk of electrocution if FJH reactions are conducted using the specified equipment without proper implementation of safety measures. The safety guidelines are listed in the Supplementary Information (Supplementary Fig. 1).

**Characterization**. SEM images were obtained using an FEI Quanta 400 ESEM FEG system at 5 kV. XRD was collected using a Rigaku D/Max Ultima II system configured with a Cu Kα radiation ($\lambda = 1.5406$ Å). XPS spectra were obtained

using a PHI Quantera XPS system under the base pressure of $5 \times 10^{-9}$ Torr. All the XPS spectra were calibrated using the standard C $1s$ peak at 284.8 eV. The temperature was measured using an infrared (IR) thermometer (Micro-Epsilon) with a temperature range of 1000–3000 °C and a time resolution of 1 ms. TGA was conducted using a Q-600 Simultaneous TGA/DSC from TA Instruments. The TGA measurement was conducted in the air with a ramp rate of 10 °C min$^{-1}$. XRF was performed using a Panalytical Axios Cement XRF. The test materials were crushed until at least 90% of the materials passed a #325 sieve (44 μm). After each sample's weight and flux amount was documented, the specimens were then transformed into glass beads by fusion via a Katanax K2 Prime instrument. Samples are heated in platinum crucibles to 1000 °C for 15 min while being rocked back and forth for dispersion. Fused lithium meta-borate/lithium tetraborate and lithium nitrate were used as fluxing agents. After fusion, the platinum crucibles containing the samples were poured into platinum molds to form beads. The fused beads were then fed into the XRF automatically via the sample loader for continued analysis. The SuperQ analytical software used the documented weights of each sample and its flux weight to generate molar quantitative results.

**Sample digestion and ICP–MS measurement of heavy metal contents.** A mixed standard was used (Millipore-Sigma, periodic table mix 1 for ICP; 33 elements; 10 mg L$^{-1}$ each; Al, As, Ba, Be, Bi, B, Ca, Cd, Cs, Cr, Co, Cu, Ga, In, Fe, Pb, Li, Mg, Mn, Ni, P, K, Rb, Se, Si, Ag, Na, Sr, S, Te, Tl, V, and Zn in 10% $HNO_3$ containing a trace of HF). $HNO_3$ (67–70 wt%, TraceMetal$^{TM}$ Grade, Fisher Chemical), HCl (37 wt%, 99.99% trace metals basis, Millipore-Sigma), $H_2O_2$ (30 wt%, for trace analysis, Millipore-Sigma), and ultrapure water (Millipore-Sigma, ACS reagent for ultratrace analysis) were used for sample digestion. The sample was digested using the method modified from a standard from the Environmental Protection Agency (EPA), USA[67]. Briefly, ~50 mg samples were added into 2 mL $HNO_3$ (67–70%, 1:1 $v:v$ with water) at 95 °C for 2 h. Then, 2 mL $H_2O_2$ (30 wt%, 1:1 $v:v$ with water) was added and heated to reflux (95 °C) for 2 h. Then, 1 mL HCl (37 wt%) and 5 mL $H_2O$ were added and heated to reflux for 15 min. The acidic solution was then filtered to remove any undissolved solid particles using a sand core funnel (Class F). The obtained solution was diluted to the range within the calibration curve, which is between 1 part per billion (ppb) to 1000 ppb. ICP-MS measurements were conducted using a PerkinElmer Nexion 300 ICP-MS system. Prior to the measurement, the ICP-MS equipment was carefully calibrated. All the samples were measured three times to obtain the standard deviation.

**Cement sample preparation and mechanical properties measurements.** The removal of residual carbon in CFA after the FJH was done by calcination at 700 °C for 1 h in the air using a furnace (NEY 6-160 A). The cement used for this project is Portland cement type I/II. Three kinds of cement specimens were cast: pure OPC, an OPC substituted with 30 wt% raw CFA, and an OPC substituted with 30 wt% purified CFA. This mass ratio (CFA:OPC = 3:7) is considered a moderate dosage of CFA for cement-based material to improve the mechanical properties without extending set time and slowing strength development[68]. All cement specimens were cast with a water:cement weight ratio of 0.6, removed from the molds after 24 h, and then cured in water for 1 day or 28 days before testing. The dimensions of the cast specimens were $25.4 \times 25.4 \times 50.8$ mm$^3$ with the shape of rectangular prisms, as shown in the inset of Fig. 4b. The cured specimens were tested on a uniaxial compressive machine with a loading rate of 1.29 mm min$^{-1}$. The load and strain were measured by the loading cell and the attached strain gauge, respectively. For each kind of sample, three specimens were measured to afford the standard deviation.

**Heavy metal leaching test of the cement samples.** Three types of cement specimens were prepared using raw CFA-C, purified CFA-C, and pure OPC with a solid mass of 0.25 g. All specimens were cast with water:cement weight ratios of 0.6 for 24 h and then cured in water for another 24 h. To mimic acid rain conditions, the specimens were separately put into a 0.0001 M $HNO_3$ solution (10 mL) with a pH of 4. The heavy metal contents in the leachate after 1, 2, 4, 6, 25, 50, 100, and 125 h were measured by ICP-MS. The accumulated heavy metal content in the leachant vs. the leaching time was plotted.

**Life cycle analysis.** The specific goal of this LCA is to evaluate the energy demands and environmental impacts resulting from different scenarios of CFA disposal or reuse in cement, some using FJH purification. The system scope considered here covers three main steps: raw material production, feedstock preparation, and landfilling. Transportation is considered here in landfill steps, and a lab-scale process is assumed for FJH with no further scaling being applied. The functional unit considered here is 1 ton of cementitious materials. A complete life cycle inventory is included in Supplementary Tables 3–7. Direct energy inputs for the FJH process were measured experimentally, and values from the ISO-compliant Argonne National Laboratory GREET LCA database or literature were used to calculate cumulative demands and impacts.

## Data availability

The data supporting the findings of this work are available within the article and its Supplementary Information. The source data generated in this study have been deposited in the Zenodo database under https://doi.org/10.5281/zenodo.7490153.

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

## Acknowledgements

The authors thank Prof. Heileen Hsu-Kim of Duke University for providing the CFA samples, and Dr. Efthymios Balomenos of MYTILINEOS for providing the BR samples. The authors thank Dr. Bo Chen of Rice University for the helpful discussion of the XPS analyses and Dr. Christopher Pennington of Rice University for developing the ICP-MS methods. The funding of the research was provided by the Air Force Office of Scientific Research (FA9550-22-1-0526, J.M.T.) and the U.S. Army Corps of Engineers, ERDC (W912HZ-21-2-0050, J.M.T.). The characterization equipment used in this project is partly from the Shared Equipment Authority (SEA) at Rice University. Permission to publish was granted by Director, Geotechnical & Structures Laboratory.

## Author contributions

B.D. conceived the idea, conducted the heavy metal removal experiments, materials characterization, LCA analysis, and wrote the manuscript. W.M. fabricated the cement samples and conducted the mechanical measurements. P.A.A. helped with the cement preparation and LCA analysis. L.E. maintained the FJH system. M.G.U.A. provided part of the CFA samples and conducted the XRF measurement. K.M.W. helped with the LCA and cement preparation. W.C. helped with the LCA. R.A.C. helped with part of the heavy metal removal work. G.L. helped with the LCA. Y.C. conducted the TGA analysis. S.N. oversaw all cement-related experiments. J.M.T. conceived the idea, wrote the manuscript, and oversaw all aspects of this article. All authors discussed the results and commented on the manuscript.

## Competing interests

Rice University owns intellectual property on the FJH strategy to remove heavy metals from CFA and BR, and the application of purified CFA in cement. An US provisional patent application was filed by Rice University (J.M.T., B.D., S.N., and M.W., Removal of Heavy Metals from Waste and Uses Thereof, US Patent App. 63/328.630), which have not yet been licensed. The authors declare no other competing interest. All other authors declare no competing interests.
