## [Peer Review File · Communications Engineering]

Reviewers' comments:

Reviewer #1 (Remarks to the Author):

In this manuscript, Deng et al. reported the heavy metal removal from coal fly ash for the application in cement. They used the flash Joule heating method which provides ultrahigh temperature to evaporate the heavy metals, leaving the main inorganic composition in fly ash. The purified coal fly ash was further used as alternative cement materials, which demonstrate better performance and lower carbon emission than traditional Portland cement. This rapid Joule heating method is novel with regard to the heavy metal decontamination from solid waste. And it is timely since the decarburization of cement industry is very urgent and difficult. The manuscript is well written. I recommended acceptance of this manuscript after the following concerns being addressed.

1. The current profile in Figure 2b is not smooth. Could the author explain why the current curve is fluctuated? Same goes to the temperature profile in Figure 2c. Is the temperature fluctuation caused by the current fluctuation?
2. In Figure 2d, the author showed the vapor pressure curves for the heavy metal. The authors should provide explanation how are these values obtained. And the curves of Co and Ni are almost overlapped. The authors should distinguish them.
3. The author should compare the flash Joule heating method with other existing methods for heavy metal removal from fly ash, such as removal efficiency, cost, scalability, etc.
4. In recent year, the ultrahigh temperature Joule heating technique has been widely used in materials synthesis and waste management. Some groundbreaking papers in this field should be cited and discussed.

Reviewer #2 (Remarks to the Author):

The authors report an effective method to remove heavy metal in the coal fly ash by FJH. In the manuscript, the authors conducted comprehensive analysis and discussions of heavy metals in the coal fly ash samples, FJH process, applications of purified coal fly ash, and techno-economics. I recommend a minor revision for the manuscript.

I do have some technical questions for the authors.

1. What is the state for the heavy metal in the coal fly ash, are they in metallic state or in some natural ore form? Will the difference effect the removal efficiency using FJH?
2. What is the optimized ratio between coal fly ash and carbon black?
3. Will minor residue of carbon impact the application of the purified sample?

Reviewer #3 (Remarks to the Author):

The manuscript titled "Heavy metal removal from coal fly ash for low carbon footprint cement" has been reviewed and it is suggested minor revision. The coal fly ash is an abundant solid waste posing a threat to the environment and conducting its treatment is significant. This work conducted a rapid and water-free process based on flash Joule heating for heavy metals removal from CFA. It is promising to produce low carbon footprint cement. It has some novelties seeking and can be accepted after some revisions.

The comments are listed as follows:

- 1) P3, 48. In some countries, the CFA is classified as hazardous waste and some are not. The

improper disposal of CFA has become an environmental concern because of potential water and soil pollution. It is not completely correct. The air pollution may also be resulted. Please refer [A comprehensive review on the applications of coal fly ash], [A review of the alumina recovery from coal fly ash, with a focus in China] and related papers.

2) Is any methods to collect these metals during FJH treatment to avoid its emission to the environment?

3) P8, line 183. The PDF card numbers should be given these crystalline phases, such as hemitate, calcite, etc.

4) Supplementary Fig.7. There are still some residue carbon in the samples after treatment. Is any other proper separation method or further studied?

Response Letter

Dear Reviewers,

We greatly thank your conscientious reviews of our manuscript, which were beneficial for strengthening our work. We answer your questions here, with point-to-point responses to them below. In this letter, the reviewers' comments are copied in *italic*; our replies are in **bold**, and changes/additions made to the manuscript and SI are in **blue** in this letter and the tracked revised manuscript and SI. If we have left any questions unanswered that was an oversight. We shall readily address that if you point out our error. Also, if you have further questions, we are eager to address those as well.

Thank you very much.

James Tour, on behalf of all the authors.

Reviewer #1 (Remarks to the Author):

In this manuscript, Deng et al. reported the heavy metal removal from coal fly ash for the application in cement. They used the flash Joule heating method which provides ultrahigh temperature to evaporate the heavy metals, leaving the main inorganic composition in fly ash. The purified coal fly ash was further used as alternative cement materials, which demonstrate better performance and lower carbon emission than traditional Portland cement. This rapid Joule heating method is novel with regard to the heavy metal decontamination from solid waste. And it is timely since the decarburization of cement industry is very urgent and difficult. The manuscript is well written. I recommended acceptance of this manuscript after the following concerns being addressed.

Response: We thank the reviewer for their support of our manuscript and their valuable comments, which are important for advancing our work. Point-by-point responses to the comments are listed below.

1. The current profile in Figure 2b is not smooth. Could the author explain why the current curve is fluctuated? Same goes to the temperature profile in Figure 2c. Is the temperature fluctuation caused by the current fluctuation?

Response: We thank the reviewer for this comment. During the FJH process, the sample resistance keeps changing because of the degassing of the sample as well as the intrinsic temperature-dependent resistivity. Hence, the current curve fluctuates, according to

Equation R1,

$$I = \frac{V}{R} \quad (\text{R1})$$

where I is the current, V is voltage, and R is sample resistance. The sample temperature (T) is dependent on the resistance and current according to the following equation,

$$Q = cmt \quad T = I^2 R t \quad (\text{R2})$$

where Q is the heat, c is the sample heat capacity, m is the sample mass, I is current, R is sample resistance, and t is the heating time. Hence, the temperature is also fluctuated during the FJH.

We revised the manuscript to explain the current and temperature fluctuations, on p 6: “The fluctuation of the current curve is ascribed to the changing of sample resistance due to degassing or intrinsic temperature-dependent resistivity.” Continuing on p 6: “The temperature keeps changing during the FJH process due to the sample resistance and current variation.”

2. In Figure 2d, the author showed the vapor pressure curves for the heavy metal. The authors should provide explanation how are these values obtained. And the curves of Co and Ni are almost overlapped. The authors should distinguish them.

Response: We thank the reviewer’s suggestion on vapor pressure curves. The vapor pressure values are standard values obtained from CRC Handbook of Chemistry and Physics.

We revised the manuscript to explain the origin of the values, in caption of Fig. 2d, “The vapor pressure values are from CRC Handbook of Chemistry and Physics⁵⁰.”

Reference added:

50. D. R. Lide, CRC Handbook of Chemistry and Physics (CRC Press, 2005)

We revised Fig. 2d to label Co and Ni:

Fig. 2d. Vapor pressure-temperature relationship of representative heavy metals and carbon.
 (This figure has replaced the original Fig. 2d in the manuscript)

3. The author should compare the flash Joule heating method with other existing methods for heavy metal removal from fly ash, such as removal efficiency, cost, scalability, etc.

Response: We thank the reviewer for the suggestions to compare our method with existing methods. The present method for heavy metal removal from CFA relies on bioleaching, acid washing (including the use of inorganic or organic acids), chemical extraction by alkaline leachates or chelating agents, and other methods. As requested by the reviewer, we compared our process with the existing processes with regards to removal efficiency, materials consumption, water consumption, materials cost, etc.

We revised the manuscript, on p 14, “We compare the FJH with existing methods⁶² for heavy metal removal from CFA (Supplementary Table 8), including bioleaching⁶³, leaching using inorganic acid⁶⁴ or organic acid⁶⁵, chemical extraction by alkaline leachates⁶⁶ or chelating agents⁶⁶. The FJH is a water-free process, while the bio- or chemical processes consume large amounts of water (Supplementary Fig. 24a). The chemical leaching methods use large amounts of chemicals such as acid, base, and chelating agents, thus the materials cost is higher than the bioleaching and FJH processes (Supplementary Fig. 24b). The FJH process affords rapid treatment within seconds, much faster than the sluggish leaching processes (Supplementary Fig. 24c). Lastly, the heavy metal removal efficiency of the FJH process resembles that of inorganic acid leaching efficiency, both of which are superior to other processes (Supplementary Fig. 24d).”

References:

62 Meer, I. & Nazir, R. Removal techniques for heavy metals from fly ash. *J. Mater. Cycles Waste Manag.* **20**, 703-722 (2018).

63 Seidel, A., Zimmels, Y. & Armon, R. Mechanism of bioleaching of coal fly ash by *Thiobacillus thiooxidans*. *Chem. Eng. J.* **83**, 123-130 (2001).

64 Xu, Y. H., Nakajima, T. & Ohki, A. Leaching of arsenic from coal fly ashes 1. leaching behavior of arsenic and mechanism study. *Toxicol. Environ. Chem.* **81**, 55-68 (2001).

65 Pangayao D, Gallardo S, editors (2014) Leaching of chromium from coal ash using citric acid, oxalic acid and gluconic acid by batch leaching procedure. In: IEEE 2014 International Conference on Humanoid, Nanotechnology, Information Technology, Communication and Control, Environment and Management (HNICEM)

66 Harris, W. R. & Silberman, D. Time-dependent leaching of coal fly ash by chelating agents. *Environ. Sci. Tech.* **17**, 139-145 (1983).

We added a table to the SI:

Supplementary Table 8. Comparison of methods for heavy metal removal from CFA.

Method	Reaction conditions	Materials Consumption per ton of CFA	Materials cost	Metals removed	Removal efficiency (%)
Bioleaching ¹²	T. thiooxidans bacteria; FAD ^a (10% w/v); 3 weeks; 28 °C; 200 rpm	Water: -9 tons Chemicals: -10 g of H ₂ SO ₄ Others: T. thiooxidans	Water: -\$9.8 Chemicals: -\$0.06 Total: -\$10	Al; Fe	25; 15-22
Inorganic acid leaching ¹³	0.1 M HNO ₃ ; L/S ^b (80-100); 24 °C; 20 h	Water: 80-100 tons Chemicals: 0.50-0.63 ton of	Water: \$86.8-108.5 Chemicals: \$9750-12285	As	83-90

		HNO ₃	Total: -\$11120		
Organic acid leaching ¹⁴	citric acid, oxalic acid and gluconic acid (0.0317 + 0.0266 + 0.0625 M); L/S (5); 30 °C; 24 h; 220 rpm	Water: 5 tons Chemicals: 30 kg of citric acid, 12 kg of oxalic acid, 61 kg of gluconic acid	Water: -\$5.4 Chemicals: -\$3472 Total: -\$3480	Cr	8.6
Alkaline leachate ¹⁵	NH ₃ (10 mM); L/S (100); 7 days; pH 7.4	Water: 100 tons Chemicals: 17 kg of NH ₃	Water: -\$108.5 Chemicals: -\$552 Total: -\$661	Mn; V; Cu; Ni; Cr;	0; 44; 7; 5; 7
Chelating agent ¹⁵	Histidine (10 mM); L/S (100); 7 days; pH 7.4	Water: 100 tons Chemicals: 155 kg of Histidine	Water: -\$108.5 Chemicals: -\$7750 Total: -\$7860	Mn; V; Cu; Ni; Cr	15; 31; 7; 4; 5
FJH, this work	Mixing with Metcoke; FJH at 60-150 V; 1 s; Separation by sieving	Water: 0 Chemicals: 25 kg of Metcoke	Water: 0 Chemicals: \$3.8 Total: \$3.8	As, Cd, Co, Ni, Pb	91; 83; 74; 69; 72

Note: ^aFAD: fly ash pulp density; ^bL/S: liquid/solid ratio; ^cMaterials prices: industrial water (\$3.10 per 100 cubic ft, or \$1.085 per ton, ref¹⁶), H₂SO₄ (\$5.8 per kg, ref¹⁷), HNO₃ (\$19.5 per kg, ref¹⁸), citric acid (\$14.3 per kg, ref¹⁹), oxalic acid (\$85.8 per kg, ref²⁰), gluconic acid (\$33 per kg, ref²¹), ammonium hydroxide 28-30% solution (\$30.9 per gallons, ref²²), Histidine (\$50 per kg, ref²³), and Metcoke (\$150 per ton, ref²⁴); ^dCFA, coal fly ash; MetCoke, metallurgical coke; FJH, flash Joule heating.

A figure has been added to the SI:

Supplementary Fig. 24. Comparison of the flash Joule heating (FJH) method with existing methods for heavy metal removal. (a) Water consumption per ton of CFA. (b) Materials cost per ton of CFA. (c) Time consumption. (d) Removal efficiencies.

4. In recent year, the ultrahigh temperature Joule heating technique has been widely used in materials synthesis and waste management. Some groundbreaking papers in this field should be cited and discussed.

Response: We thank the reviewer for their comment on the background. The Joule heating technique was first used for materials synthesis by Prof. Liangbing Hu's group at University of Maryland (*Nat. Commun.* 2016, 7, 12332) for the synthesis of nanoparticles supported on

reduced graphene oxide films. Later, Prof. Hu's group extended this method for various alloy nanoparticle synthesis, ceramic sintering, single-atom catalysts, and others. Our group first developed the flash Joule heating for converting various carbon-containing sources into flash graphene (*Nature* 2020, 577, 647). Our group further extended the FJH process for waste management, including urban mining of precious metals from electronic waste, rare earth recovery from secondary wastes, sustainable management of organic wastes especially plastic, and other applications. Inspired by Prof. Hu and our group's work, the Joule heating techniques for materials synthesis and waste management are rapidly growing. We agree with the reviewer that more groundbreaking papers in this field should be cited and discussed.

We have revised our manuscript accordingly, on p 3, "Chen et al. first reported the rapid Joule heating for ultrafast synthesis of nanoparticles in reduced graphene oxide films²⁷. The carbothermic shock was then widely applied for various nanomaterials syntheses³⁰, including silicon nanoparticles³¹, high-entropy alloy nanoparticles²⁸, and single-atom catalysts³².", continuing p 3, "end-of-life vehicle waste⁴⁰, and asphaltenes⁴¹", and on p 4, "the recycling of photovoltaic silicon waste⁴⁴, and recovery of lithium-ion batteries anodes^{45,46} and cathodes⁴⁷."

References

- 27 Chen, Y. *et al.* Ultra-fast self-assembly and stabilization of reactive nanoparticles in reduced graphene oxide films. *Nat. Commun.* 7, 12332 (2016).
- 30 Jiang, R. *et al.* Ultrafast synthesis for functional nanomaterials. *Cell Rep. Phys. Sci.* 2, 100302 (2021).
- 31 Chen, Y. *et al.* Rapid, in situ synthesis of high capacity battery anodes through high temperature radiation-based thermal shock. *Nano Lett.* 16, 5553-5558 (2016).
- 32 Yao, Y. *et al.* High temperature shockwave stabilized single atoms. *Nat. Nanotechnol.* 14, 851-857 (2019).
- 40 Wyss, K. M. *et al.* Upcycling end-of-life vehicle waste plastic into flash graphene. *Commun. Eng.* 1, 3 (2022).
- 41 Saadi, M. A. S. R. *et al.* Sustainable valorization of asphaltenes via flash joule heating. *Sci. Adv.* 8, eadd3555 (2022).
- 44 Lu, J. *et al.* Millisecond conversion of photovoltaic silicon waste to binder-free high silicon content nanowires electrodes. *Adv. Energy Mater.* 11, 2102103 (2021).
- 45 Dong, S. *et al.* Ultra-fast, low-cost, and green regeneration of graphite anode using flash joule heating method. *EcoMat* 4, e12212 (2022).
- 46 Chen, W. *et al.* Flash recycling of graphite anodes. *Adv. Mater.* doi: 10.1002/adma.202207303 (2022).
- 47 Cui, B. *et al.* Waste to wealth: Defect-rich Ni-incorporated spent LiFePO₄ for efficient oxygen evolution reaction. *Sci. China Mater.* 64, 2710-2718 (2021).

Reviewer #2 (Remarks to the Author):

The authors report an effective method to remove heavy metal in the coal fly ash by FJH. In the manuscript, the authors conducted comprehensive analysis and discussions of heavy metals in the coal fly ash samples, FJH process, applications of purified coal fly ash, and techno-economics. I recommend a minor revision for the manuscript.

Response: We thank the reviewer for the positive evaluation to our work.

I do have some technical questions for the authors.

1. What is the state for the heavy metal in the coal fly ash, are they in metallic state or in some natural ore form? Will the difference effect the removal efficiency using FJH?

Response: We thank the reviewer for this question. The heavy metals in coal fly ash are trace metals so the chemical state cannot be analyzed using traditional XPS due to the detection limits. The chemical state of the trace metals in coal fly ash can be analyzed by X-ray absorption spectroscopy (XAS) using synchrotron light, which has been extensively previously studied. For example, Rivera *et al.* found that the majority of As is oxidized As(V) associated with iron phases or aluminosilicate glass; Se is mostly Se(IV) with lesser Se(VI); Zn is Zn(II) with ferrihydrite, franklinite ($ZnFe_2O_4$) and ZnO; and Cr is oxidized Cr(III) (*Energy Fuels* 2017, 31, 9652). For Pb, since the Pb-bearing sulfides thermally decompose during the coal combustion process, Pb is primarily transformed into $PbCl_2$, PbO and $PbSO_4$ (*Fuel Process Technol* 2011, 92, 441). Liu *et al.* reported that Ni is mainly in Fe oxides and as NiO, and Cu mainly as Cu_2O/CuO or in Fe oxides (*Energy Fuels* 2020, 34, 14333). Hence, the heavy metals in coal fly ash are in natural ore forms or their oxides. This is reasonable since coal combustion is in a high temperature and oxidation atmosphere.

The differences between the metallic state and oxide state might affect the removal efficiency. In the FJH process, the heavy metals are removed by evaporative separation, so the difference of vapor pressure of the heavy metal species would affect the removal efficiencies. The heavy metals in CFA are in natural ore or oxide form. Depending on the reactivity and thermal stability of these species, there are several scenarios: (1) The heavy metal species evaporate directly; (2) The heavy metal species thermally decompose to other compounds and then evaporate; (3) The heavy metal compounds are carbothermically reduced to elemental metals and then evaporate. Fortunately, FJH can achieve an ultrahigh temperature up to 3000 °C, which is higher than the temperature required for all of the above scenarios. We illustrate this by using the different speciation of lead (Pb, $PbCl_2$, PbO, PbS, and $PbSO_4$) as an example for the thermodynamic analysis. Pb and $PbCl_2$ could directly evaporate. PbO could directly evaporate, or it can be carbothermically reduced to Pb(0) by the reaction:

PbS could be converted to PbO by the reaction:

) PbSO₄ could decompose to PbO by the reaction:

We calculated the Gibbs free energy change of these reaction (Fig. R2a) using HSC Chemistry 10 software. All these reactions are thermodynamically favorable below 2000 °C. Furthermore, we calculated the vapor pressure as it varied with temperature for different Pb species (Fig. R2b). All these Pb species have a vapor pressure below 2000 °C. Since our FJH process can achieve a high temperature of 3000 °C, all the above chemical conversion and evaporation processes are thermodynamically favorable and rapid. Based on this, we conclude that the FJH process is applicable to heavy metals removal regardless of their chemical states.

We revised the manuscript on p 7: “The heavy metals in CFA are present in the oxidized or natural ore forms, according to previous studies^{10,53,54}. Depending on the reactivity and thermal stability, the heavy metal species could evaporate in the natural form, or they could undergo thermal decomposition or carbothermic reduction to other compounds or elemental metals and then evaporate. In any case, the ultrahigh temperature by the FJH process would enable the chemical conversion and evaporation of the heavy metal species, which usually take place far below 3000 °C (Supplementary Fig. 5).”

A figure has been added to the SI:

Supplementary Fig. 5. Effect of chemical state to the FJH process. (a) Thermodynamic analysis of carbothermic reduction of PbO and thermal decomposition of PbSO₄. (b) Vapor pressure-temperature relationship of Pb species.

(This figure has been added to SI as Supplementary Fig. 5)

References:

- 53 Rivera, N., Hesterberg, D., Kaur, N. & Duckworth, O. W. Chemical speciation of potentially toxic trace metals in coal fly ash associated with the Kingston fly ash spill. *Energy Fuels* 31, 9652-9659 (2017).
- 54 Liu, P., Wang, Q., Jung, H. & Tang, Y. Speciation, distribution, and mobility of hazardous trace elements in coal fly ash: Insights from Cr, Ni, and Cu. *Energy Fuels* 34, 14333-14343 (2020).

2. What is the optimized ratio between coal fly ash and carbon black?

Response: The carbon additives carbon black, metallurgical coke, plastic pyrolysis ash, and other carbon-containing materials are used as conductive additives to afford appropriate resistance for Joule heating. In the FJH process, the sample resistance is important: if the resistance is too high, the current is not large enough to generate heating; in contrast, if the resistance is too low, the sample resembles a conductor and cannot generate enough heat. Based on our experience, a resistance of $\sim 1 \Omega$ is the best resistance for the FJH process. From the aspect of materials cost, we aim to use the least carbon additive possible. The measured resistance varied with the mass ratio of CFA and carbon black (Fig. R3). The mass ratio of $m(\text{CFA}):m(\text{CB}) = 2:1$ is an optimized value with $R \sim 1 \Omega$. As we have shown in the manuscript, the carbon additives can be recovered and reused, so the materials cost would be minimized in that case.

We revised the manuscript accordingly on p 7, “The optimized mass ratio of CFA and CB was $\sim 2:1$ (Supplementary Table 1), where the resistance of the sample was $\sim 1 \Omega$ (Supplementary Fig. 6).”

Supplementary Fig. 6. Sample resistance measurement. Sample resistance varied with mass ratio of coal fly ash (CFA) and carbon black (CB). The dashed line indicates $R = 1 \Omega$.

(This figure has been added into the SI as Supplementary Fig. 6)

3. Will minor residue of carbon impact the application of the purified sample?

Response: We thank the reviewer for this question regarding carbon residues. As shown in our manuscript, there are two processes to remove carbon residues after the FJH process. The first one is by calcination in air (Fig. 4a), wherein the carbon can be almost completely removed; but this process is energy intensive. The second one is by sieving (Supplementary Fig. 9), that leaves behind minor carbon residues. We here measured the carbon content using thermogravimetric analysis (TGA) after the sieving process, which is ~3% (Fig. R4). Actually, the class F CFA itself contains some carbon residues (Fig. 1b). For the applications that are tolerant to minor carbon, we can use the sieving process; for the applications that are susceptible to minor carbon, we can use the calcination process. Specifically, for the application in cement, our previous publications have demonstrated that minor carbon content actually enhances the cement performances (Luong, et al. *Nature* 2020, 577, 647; Advincula, et al. *Carbon* 2021, 178, 649).

We revised our manuscript accordingly, on p 8, “After the sieving separation process, the residual carbon content in the treated CFA was reduced to ~3% (Supplementary Fig. 10). The residual carbon could be almost completely removed by calcination in air, which will be discussed later. The choice of appropriate carbon removal approaches would depend on the landfilling or applications of the purified CFA.”

Supplementary Fig. 10. Residual carbon in the CFA sample after sieving separation. TGA was conducted in air with the heating rate of 10 °C/min.

This figure has been added into the SI as Supplementary Fig. 10.

Reviewer #3 (Remarks to the Author):

The manuscript titled "Heavy metal removal from coal fly ash for low carbon footprint cement" has been reviewed and it is suggested minor revision. The coal fly ash is an abundant solid waste posing a threat to the environment and conducting its treatment is significant. This work conducted a rapid and water-free process based on flash Joule heating for heavy metals removal from CFA. It is promising to produce low carbon footprint cement. It has some novelties seeking and can be accepted after some revisions.

Response: We thank the reviewer for the positive evaluation to our work.

The comments are listed as follows:

1) P3, 48. In some countries, the CFA is classified as hazardous waste and some are not. The improper disposal of CFA has become an environmental concern because of potential water and soil pollution. It is not completely correct. The air pollution may also be resulted. Please refer [A comprehensive review on the applications of coal fly ash], [A review of the alumina recovery from coal fly ash, with a focus in China] and related papers.

Response: We thank the reviewer for their comment on the research background. We agree with the reviewer that CFA is classified as hazardous wastes in some countries but not in others. For example, the US EPA defines coal fly ash as a “special waste” to avoid the stringent hazardous waste permitting requirement, but some regulations are still required. We also agreed that the improper disposal of CFA could cause air pollution as well.

We revised the manuscript, on p 3: “CFA is classified as hazardous waste in some countries if the toxic substances exceed limits; other countries regulate it as non-hazardous with special regulations¹¹. The improper disposal of CFA has become an environmental concern because of potential water, soil, and air pollution^{6,12}.”

References:

- 6 Yao, Z. T. *et al.* A comprehensive review on the applications of coal fly ash. *Earth-Sci. Rev.* 141, 105-121 (2015).
- 11 US EPA. "Hazardous and Solid Waste Management System; Disposal of Coal Combustion Residuals From Electric Utilities; Final rule." *Federal Register*, 80 FR 21301, 2015-04-17.
- 12 Yao, Z. T., Xia, M. S., Sarker, P. K. & Chen, T. A review of the alumina recovery from coal fly ash, with a focus in China. *Fuel* 120, 74-85 (2014).

2) Is any methods to collect these metals during FJH treatment to avoid its emission to the environment?

Response: In our current bench-scale experiment, the coal fly ash samples were loaded into a quartz tube (Supplementary Fig. 1b), which was placed into a sealed chamber

(Supplementary Fig. 1d). The evaporated heavy metals were deposited on the quartz tube side wall or inside the vacuum chamber, instead of emission to the environment.

We revised the manuscript to make this point clearer, on p 7, “The evaporated heavy metals were deposited onto the sidewall of the quartz tube reactor or inside the sealed chamber (Supplementary Figs. 1b,d), in avoidance of emission to the environment.”

3) P8, line 183. The PDF card numbers should be given these crystalline phases, such as hematite, calcite, etc.

Response: We thank the reviewer for this suggestion. We have added the PDF card numbers to all the XRD patterns.

Caption of Fig. 1a:

“Mullite (PDF 15-0776) and quartz (PDF 33-1161) are used as references.”

Caption of Fig. 3c:

“Hematite (PDF 02-0919) and calcite (PDF 47-1743) are used as references.”

Caption of Supplementary Fig. 11:

“Mullite (PDF 15-0776) and quartz (PDF 33-1161) are used as references.”

Caption of Supplementary Fig. 14:

“Mullite (PDF 15-0776) and quartz (PDF 33-1161) are used as references.”

Caption of Supplementary Fig. 15:

“Hematite (PDF 02-0919) and calcite (PDF 47-1743) are used as references.”

4) *Supplementary Fig.7. There are still some residue carbon in the samples after treatment. Is any other proper separation method or further studied?*

Response: In this work, we proposed two protocols to remove the carbon after the FJH process. The first one is by calcination in air at 700 °C for 1 h (Fig. 4a), wherein the carbon can be almost completely removed. The second one is by sieving (original Supplementary Fig. 7, now Supplementary Fig. 9), wherein there are minor carbon residue, ~3% according to TGA (Fig. R4). For the landfilling and some applications that are tolerant to minor carbon residues, the sieving process can be used because it is faster and less energy consumptive. In contrast, for the applications of the purified coal fly ash that are sensitive to minor carbon residue, we may use the calcination process.

We revised our manuscript accordingly, on p 8, “After the sieving separation process, the residual carbon content in the treated CFA was reduced to ~3% (Supplementary Fig. 10).

The residual carbon could be almost completely removed by calcination in air, which will be discussed later. The choice of appropriate carbon removal approaches would depend on the landfilling or applications of the purified CFA.”

Supplementary Fig. 10. Residual carbon in the CFA sample after sieving separation. TGA was conducted in air with the heating rate of 10 °C/min.

This figure has been added into the SI as Supplementary Fig. 10.

REVIEWERS' COMMENTS:

Reviewer #1 (Remarks to the Author):

The authors have addressed my comments. The manuscript is in a good shape to publish now. In addition, about the "recovery of lithium-ion batteries anodes", another recently published reference is also suggested to be cited: [10.1007/s12274-022-5244-z](https://doi.org/10.1007/s12274-022-5244-z).

Reviewer #3 (Remarks to the Author):

It has been revised according to the comments and can be accepted.

Response Letter

Note: The reviewers' comments are copied in *italics*; our response is regular; the revised content is marked in blue.

Reviewer #1:

The authors have addressed my comments. The manuscript is in a good shape to publish now. In addition, about the "recovery of lithium-ion batteries anodes", another recently published reference is also suggested to be cited: 10.1007/s12274-022-5244-z.

Response: We thank the reviewer for this suggestion. The reference mentioned by the reviewer is about the recycling of spent battery graphite anode by the high-temperature shock process, which is definitely related to the topic of waste recycling by rapid electrothermal process.

We added the reference in the revised manuscript, on Page 4, "and recovery of lithium-ion batteries graphite anodes⁴⁵⁻⁴⁷".

Reference:

47. Luo, J. et al. Recycle spent graphite to defect-engineered, high-power graphite anode. *Nano Res.*, doi:10.1007/s12274-022-5244-z (2022).

Reviewer #3:

It has been revised according to the comments and can be accepted.

Response: We thank the reviewer.